# Lactofermented Annurca Apple Puree as a Functional Food Indicated for the Control of Plasma Lipid and Oxidative Amine Levels: Results from a Randomised Clinical Trial

**DOI:** 10.3390/nu11010122

**Published:** 2019-01-09

**Authors:** Gian Carlo Tenore, Domenico Caruso, Giuseppe Buonomo, Maria D’Avino, Roberto Ciampaglia, Maria Maisto, Connie Schisano, Bruno Bocchino, Ettore Novellino

**Affiliations:** 1Department of Pharmacy, University of Naples Federico II, Via D. Montesano 49, 80131 Naples, Italy; rociamp@unina.it (R.C.); marmais@unina.it (M.M.); conschi@unina.it (C.S.); enov@unina.it (E.N.); 2Department of Internal Medicine, Hospital Cardarelli, Via Antonio Cardarelli, 80131 Naples, Italy; domcarus@gmail.com (D.C.); maridav@gmail.com (M.D.); 3Coop. Samnium Medica, Viale C. Colombo, 18, 82037 Benevento, Italy; giusbuon@tin.it; 4UCCP (Unità Complessa Cure Primarie), Via Manzoni, San Giorgio del Sannio, 82100 Benevento, Italy; brubocch@email.it

**Keywords:** functional food, Annurca apple puree, *Lactobacillus*, fermentation, plasma lipids, trimethylamine-N-oxide

## Abstract

Atherosclerotic cardiovascular diseases are preferential targets of healthy diet and preventive medicine partially through strategies to improve lipid profile and counteract oxidative metabolites. Ninety individuals with cardiovascular disease (CVD) risk factors were randomized (1:1:1 ratio) to three arms, according to a four-week run-in, eight-week intervention, and four-week follow up study, testing the effects of a lactofermented Annurca apple puree (lfAAP), compared to unfermented apple puree (AAP) or probiotic alone (LAB) on plasma lipid profile and trimethylamine-N-oxide (TMAO) levels. By comparing the treatments, data indicated for the subjects tested with lfAAP a higher variation of the following serum parameters, in respect to the other treatment groups: high density lipoprotein cholesterol (HDL-C), +61.8% (*p* = 0.0095); and TMAO levels, −63.1% (*p* = 0.0042). The present study would suggest lfAAP as an effective functional food for beneficial control of plasma HDL-C and TMAO levels.

## 1. Introduction

Polyphenols are a major group of bioactive substances found in fruits and vegetables with potential health benefits. As polyphenols characteristically have radical-scavenging activities, they exhibit antioxidant, as well as anti-inflammatory and anticarcinogenic behaviours [1,2]. Epidemiological studies show that dietary consumption of foods containing these compounds can result in a decreased risk of the development of certain lifestyle-related chronic diseases such as cardiovascular disease and colorectal cancer [2,3].

In order for polyphenols in fruits and vegetables to be available for absorption (i.e., become bioavailable) within the human gastrointestinal tract, these compounds must be released from the plant cell (i.e., be bioaccessible). When cells break during processing or oral mastication of fruits and vegetables, lipid membranes and the cellulose-pectin based plant cell wall (PCW) are fractured, allowing the contents within the cell to be released [4,5]. During exit from the cell, polyphenols come into contact with PCW for the first time, with the potential for binding interactions to take place, before they become available for uptake. Any binding interactions would be expected to be relevant for processed forms of fruits and vegetables, such as purees and sauces, where cell walls are deliberately ruptured during processing, thereby allowing polyphenol–PCW interactions to occur over a longer time [4,5]. Previous authors have demonstrated that specific apple polyphenols, the oligomeric procyanidins, are capable of selective interactions with individual cell wall constituents, with the neutral procyanidins interacting more extensively with pectin than cellulose [6,7]. Consequently, interactions between different polyphenols and the PCW, are likely to affect intestinal bioaccessibility of fruit and vegetable polyphenols. At first sight, non-bioaccessible polyphenols (e.g., by being bound to PCW material) would appear desirable, since they would survive to the large intestine, thus, being important for maintaining good gut health. In fact, epidemiological studies show that diets high in fruits and vegetables are associated with lower rates of colorectal cancer [8]. However, bioaccessible polyphenols represent the only form of antioxidant compounds, which being absorbed by the small intestine, can reach the systemic circulation and exert protective effects at the cardiovascular level [2]. 

Over the last decades, fermented foods are the object of a renewed interest in Western countries [9]. Mainly of vegetable origin, these products are strongly appreciated for several reasons, primarily nutrition–health approaches, followed by food safety, desirable organoleptic modification, shelf life extension, simplicity of preparation, valorisation of unused raw vegetal material, and sustainable development [10,11,12]. Among the wide range of microbial strains responsible for the fermentation of vegetable and fruits, in many cases lactic acid bacteria (LAB) are involved. *Lactobacillus* species are among the most frequent species of LAB involved in fruits or vegetables fermentation, specifically, *L. plantarum*, *L. brevis*, *L. rhamnosus*, and *L. acidophilus* [13]. Lactic fermentation modifies the composition of the fermented materials and can improve the beneficial health benefits of food by several mechanisms. Of major interest, it is reported that lactic fermentation can enhance the intestinal bioaccessibility of polyphenols through hydrolysis of esters with PCW polymeric constituents. Specifically, some authors have demonstrated that the decrease in soluble and insoluble fibre during fermentation processes in vegetables could be attributed to the hydrolysis of pectic compounds and the use of cellulose and hemicellulose as substrates by the microorganisms responsible for the fermentation [14,15,16].

Our research group has previously evaluated the polyphenolic composition of Annurca apples, the only apple cultivar native to the Campania region (Southern Italy), listed as a Protected Geographical Indication (PGI) product (Commission Regulation (EC) No. 417/2006)) [17]. By comparing experimental data with those from more common commercial apple cultivars (i.e., Red Delicious, Pink Lady, Fuji, Golden Delicious), Annurca apples exhibited the highest polyphenolic concentration, especially as regards oligomeric procyanidins. Moreover, our previous clinical study has demonstrated that a daily consumption of Annurca apples beneficially affects the serum cholesterol levels in healthy subjects, providing the most appreciable results among different apple cultivars tested [18]. In the light of what is stated above, it can be hypothesised that lactic acid fermentation may play a major role in enhancing the potential functional properties of this food product. Thus, the aims of the present work were to: i) evaluate the effects of lactic acid fermentation on the levels of free polyphenolic compounds in Annurca apple puree; and ii) compare the effects of lactofermented Annurca apple puree, unfermented Annurca apple puree, and probiotic alone, on human plasma lipid profile and oxidative amines.

## 2. Materials and Methods

### 2.1. Reagents and Standards

All chemicals and reagents used were either analytical-reagent or HPLC grade. The water was treated in a Milli-Q water purification system (Millipore, Bedford, MA, USA) before use. The standards used for the identification and quantification of phenolic acids and flavonoids were: chlorogenic acid, (+)-catechin, (−)-epicatechin, isorhamnetin, myricetin, phloretin, phloridzin (phloretin-2-O-glucoside), procyanidin B2, quercetin, quercitrin (quercetin-3-O-rhamnoside), rutin (quercetin-3-O-rutinoside), isoquercitrin (quercetin-3-O-glucoside), hyperin (quercetin-3-O-galactoside), and cyanidin-3-O-galactoside chloride (Sigma Chemical Co., St. Louis, MO, USA). Acetonitrile and methyl alcohol were of HPLC grade (Carlo Erba, Milano, Italy). Sodium taurocholate, phosphatidylcholine, cholesterol, NaCl, and sodium phosphate were purchased from Sigma Chemical Co.

### 2.2. Fruit Collection and Annurca Apple Puree (AAP) Preparation

Annurca (*M. pumila* Miller cv Annurca) apple fruits were collected in Valle di Maddaloni (Caserta, Italy) in October 2016 when fruits had just been harvested (green peel). Fruits were reddened, following the typical treatment for about 30 days, and then analysed [19]. Large-scale production of AAP was accomplished by Masseria Giòsole (Caserta, Italy). Apples were milled with hot (90 °C) water to a fine puree which was hermetically closed into sterile glass jars and submitted to sterilisation (120 °C for 15 min).

### 2.3. Lactofermented AAP (lfAAP) Preparation

Inoculation of AAP with the probiotic strains *Lactobacillus rhamnosus* LRH11 and *Lactobacillus plantarum* SGL07 (Farmalabor, Canosa, Italy) was performed as previously reported [20,21] with slight modifications. Probiotic strains were reactivated by culturing twice in MRS broth (meat peptone, 10.0 g/L; dextrose, 20.0 g/L; yeast extract, 5.0 g/L; beef extract 10.0 g/L; disodium phosphate 2.0 g/L; sodium acetate, 5.0 g/L; ammonium citrate, 2.0 g/L; magnesium sulphate 0.1 g/L; manganese sulphate 0.05 g/L; Tween 80 1.0 g/L) (Thermo scientific, Whalthan, MA, USA) at 37 °C for 18 h. Pellets were washed in sterile physiological solution (NaCl 8.5 g/L) and suspended in 5 mL of the same solution. The sterilised jars containing AAP were opened in sterile conditions and inoculated with one or the other probiotic strain to obtain a bacterial load of about 2.5 × 10^6^ CFU/g. Both types of samples were incubated at room temperature for 48 h and stored at 4 °C for 4 weeks. At each of the three steps of fermentation (0–24–48 h), aliquots of the lactofermented AAP (lfAAP) were collected for microbiological, chemical-physical, and chemical analyses. 

### 2.4. Enumeration of Probiotic Microorganisms in lfAAP

Viability of probiotic cultures in lfAAP was determined and expressed as colony forming units (CFU)/mL on MRS agar (Oxoid, Milan, Italy). Serial dilutions were prepared in sterile physiological solution before plating onto MRS agar. Plates were incubated at 37 °C for 48 h in an Oxoid anaerobic system.

### 2.5. Preparation of Polyphenolic Extracts from AAP and lfAAP Samples

Lyophilized samples (10 g) were treated with 100 mL of 80% methanol (0.5% formic acid) for 24 h at 4 °C to extract phenolic compounds. After centrifugation, the supernatant was slowly filtered through an Amberlite XAD-2 column packed as follows: resin (10 g; pore size 9 nm; particle size 0.3–1.2 mm; Supelco, Bellefonte, PA, USA) was soaked in methanol, stirred for 10 min and then packed into a glass column (10 × 2 cm). The column was washed with 100 mL of acidified water (pH 2) and 50 mL of deionised water for sugar, and other polar compound removal. The adsorbed phenolic compounds were extracted from the resin by elution with 100 mL of methanol, which was evaporated by flushing with nitrogen. Samples were stored at −20 °C until HPLC analysis.

### 2.6. HPLC-DAD/ESI-MS Analysis of Polyphenolic Extracts

Polyphenolic extracts were solubilized with 1% formic acid. Analyses were run on a Jasco Extrema LC-4000 system (Jasco Inc., Easton, MD, USA) provided with photodiode array detector (DAD). The column selected was a Kinetex^®^ C18 column (250 mm × 4.6 mm, 5 μm; Phenomenex, Torrance, CA, USA). The analyses were performed at a flow rate of 1 mL/min, with solvent A (2% acetic acid) and solvent B (0.5% acetic acid in acetonitrile and water 50:50, v/v). After a 5 min hold at 10% solvent B, elution was performed according to the following conditions: from 10% (B) to 55% (B) in 50 min and to 95% (B) in 10 min, followed by 5 min of maintenance. Flavonols, procyanidins, dihydrochalcones, flavanols, and hydroxycinnamic acids were monitored at 280 nm and anthocyanins at 520 nm. For quantitative analysis, standard curves for each polyphenol standard were prepared over a concentration range of 0.1–1.0 μg/μL with six different concentration levels and duplicate injections at each level. The identity of polyphenols was confirmed by LC-ESI/MS experiments and data were compared with those of commercial standards. The same chromatographic apparatus and conditions (HPLC system, gradient elution, column, temperature) were coupled to an Advion Expression mass spectrometer (Advion Inc., Ithaca, NY, USA) equipped with an electrospray (ESI) source. Mass spectra were recorded from m/z = 50 to 1200, both in negative and in positive ionization mode. The capillary voltage was set at −28 V, the spray voltage was at 3 kV, and the tube lens offset was at −10 V in negative ion mode, while the capillary voltage was set at 34 V, the spray voltage was at 3.5 kV, and the tube lens offset was at 55 V in positive ion mode. The capillary temperature was 275 °C. Data were acquired in full scan and SIM modes.

### 2.7. Study Population and Protocol

Study participants were recruited by the Samnium Medical Cooperative (Benevento, Italy). Subjects were enrolled in January 2017. Subjects aged 18–70 years were eligible for enrolment if they had the following values of serum parameters at baseline: total cholesterol (TC), 200–260 mg/dL; HDL-C, 30–45 mg/dL; LDL-C, 150–182 mg/dL; glucose, 90–125 mg/dL; TG, 170–280 mg/dL. 

Exclusion criteria were: smoking, obesity (BMI > 30 kg/m^2^), diabetes, serious hepatic disease (cirrhosis, hepatitis), serious renal disorders (serum creatinine > 2.8 mg/dL), heart disease, family history of chronic diseases, drug therapy or supplement intake for hypercholesterolemia, drug therapy or supplement intake containing apple polyphenols, heavy physical exercise (>10 h/week), pregnant women, women suspected of being pregnant, women who hoped to become pregnant, breastfeeding, birch pollen allergy, use of vitamin/mineral supplements 2 weeks prior to entry into the study, and donation of blood less than 3 months before the study. 

The subjects received oral and written information concerning the study before they gave their written consent. Protocol, letter of intent of volunteers, and synoptic document about the study were submitted to the Scientific Ethics Committee of AO Rummo Hospital (Benevento, Italy). The study was approved by the committee (protocol 10/2016/PR of 22/10/2016) and carried out in accordance with the Helsinki declaration of 1964 (as revised in 2000). The subjects were asked to make records in an intake-checking table for the intervention study and side effects in daily reports. The study was a randomised, single centre trial conducted at the Samnium Medical Cooperative (Benevento, Italy). 

The study duration was 16 weeks. The subjects were randomly divided into three groups (AAP, lfAAP, and LAB). Each group underwent 4 weeks of run-in period (no treatment), followed by 8 weeks of intervention, and 4 weeks of follow-up. All of the subjects were provided all 8-weeks of the treatment doses at randomization. During the intervention periods, subjects were administered according to the belonging group, as follows: AAP group, Annurca apple puree (125 g/day); lfAAP group, lactofermented Annurca apple puree (125 g/day); LAB group, 1 capsule/day, containing the exact amount of *Lactobacillus* strain occurring in 125 g of lfAAP (about, 3.0 × 10^8^ CFU). Subjects were instructed to take AAP, lfAAP, or LAB, at one of the meals. Subjects were asked to keep their dietary habits unchanged throughout the entire study. To this regard, they were provided with a food diary on which annotate their daily dietary habits and were instructed to maintain their habitual patterns of physical activity throughout the entire study period. 

Both the examinations and the study treatment were performed in an outpatient setting. Clinic visits, and blood and faecal swab sampling were performed after 12 h of fasting at weeks 0, 4, 8, 12, and 16. Subjects were informed not to drink alcohol or perform hard physical activity 48 h prior to blood sampling. All blood samples were taken in the morning and immediately after measurement of heart rate and blood pressure. Blood samples were collected in 10-mL EDTA-coated tubes (Becton–Dickinson, Plymouth, UK) and plasma was isolated by centrifugation (20 min, 2.200 *g*, 4 °C). All samples were stored at −80 °C until analysis. 

Plasma TC, HDL-C, LDL-C, glucose, and TG levels were determined using commercially available kits from Diacron International (Grosseto, Italy). Analyses were performed on a Diacron International Free Carpe Diem. The assay sensitivity for the individual analytical determination was determined as follows: TC, 1 mg/dL; HDL-C, 1 mg/dL; LDL-C, 5 mg/dL; glucose, 4 mg/dL; TG, 3 mg/dL. Intra- and inter-day variations were 1.4% and 1.6% for TC, 1.6% and 2.2% for LDL-C, 2.0% and 2.3% for HDL-C, 1.1% and 1.7% for glucose, and 1.3% and 1.8% for TG, respectively. 

Trimethylamine-N-oxide (TMAO) was quantified in blood samples as previously reported [22], with slight modifications. An aliquot of 80 µL of plasma was added to 160 µL of methanol. The mixture was vortexed for 2 min, and then centrifuged at 16.128 rcf. The supernatant was collected and analysed by HPLC/ESI-MS technique. The HPLC system Jasco Extrema LC-4000 system (Jasco Inc., Easton, MD, USA) was coupled to an Advion Expression mass spectrometer (Advion Inc., Ithaca, NY, USA). An electrospray ionization (ESI) source was used in positive ion mode. Source and capillary temperatures were 150 and 400 °C, respectively. Capillary voltage was +0.60 kV, and desolvation and cone gas (both N_2_) flow rates were 800 and 20 L/h, respectively. For the separation of the analytes, a Luna Hilic (5µ particle size, 150 × 3 mm) and security guard colon both supplied by Phenomenex (Torrance, CA, USA,) were used. The column temperature was maintained at 60 °C during analysis. Mobile phase composition comprised (A) 0.15% formic acid in water containing a final concentration of 10 mM ammonium acetate, and (B) 100% methanol (LC/MS grade) in the ratio 80:20 (A:B), run isocratically at a flow rate of 0.35 mL/min for 6 min, with a 5 µl injection volume. After use, the column was stored in 100% acetonitrile and was routinely cleaned according to the manufacturer’s instructions. Trimethylamine-N-oxide was identified and quantified by analytical standard (Sigma–Aldrich St. Louis, MO, USA) and calibration curve.

Faecal swabs were analysed for their microbial composition. Specifically, the following BD (Becton Dickinson, Heidelberg, Germany) ready-for-use culture plates were purchased and employed according to instructions: *Bifidobacterium* agar, modified; *Bacteroides* bile esculin agar with amikacin; *Lactobacillus* (LBS) selection agar; *Enterococcus* agar.

In addition to the five clinic visits, six standardised telephone interviews were performed every 14 days starting from the first clinic visit, to verify compliance and increase protocol adherence. In particular, these interviews reminded subjects to complete their intake-checking table for the intervention study and to record any treatment discontinuation, or adverse events they might have experienced in the meantime (which were also documented regularly on the case report forms during each telephone and clinic visit).

All subjects underwent a standardised physical examination, assessment of medical history (up to five years before enrolment), laboratory examination, measurement of blood pressure and heart rate, and evaluation of Body mass index (BMI). Body mass index was calculated from body height and body weight. Body fat percentage was measured using a body composition analyser (TBF-310, Tanita Corp., Tokyo, Japan) and systolic blood pressure, diastolic blood pressure, and heart rate were measured using an HBP-9020 (OMRON COLIN Corp., Tokyo, Japan). At each clinic visit, subjects had to complete three self-administered questionnaires on quality of life aspects, and their diaries were checked for data completeness and quality of documentation to ensure subject comprehension of the diary items. 

### 2.8. Randomisation, Concealment, and Blinding

A total of 90 eligible subjects (51 men and 39 women, 18–70 years of age) were randomly assigned to three sub-groups (each one of 30 subjects). If a subject dropped out before the intervention period, he or she was replaced by the next eligible subject enrolled at the same centre. The concealed allocation was performed by an internet-based randomisation schedule, stratified by study site. The random number list was generated by an investigator with no clinical involvement in the trial. Subjects, clinicians, core laboratories, and trial staff (data analysts, statisticians) were blind to treatment allocation. 

### 2.9. Study Outcomes and Data Collection

#### 2.9.1. Primary and Secondary Efficacy Outcomes

Primary endpoints measured were plasma level variations of TC, HDL-C, LDL-C, glucose, TG, and TMAO, while key secondary outcomes were microbial composition of faecal swabs and parameters collected during clinic visits, such as blood pressure, heart rate, and evaluation of BMI. All raw subject ratings were evaluated in a blinded manner at the site of the principal investigator. 

#### 2.9.2. Safety

We assessed safety from reports of adverse events as well as laboratory parameters concerning the hepatic and renal function, vital signs (i.e., blood pressure, pulse, height, weight, and body mass index), and physical or neurological examinations. Safety was assessed over the entire treatment period at weeks 0, 4, 8, 12, and 16, including adverse events occurring in the first three weeks after cessation of treatments.

### 2.10. Statistics

#### 2.10.1. Methodology

During the trial, it became apparent that dropouts and incomplete diary documentation created missing data that could not be adequately handled by the intended robust comparison. To deal with the missing data structure, we used a negative binomial, generalised linear mixed effects model (NB GLMM) that not only yields unbiased parameter estimates when missing observations are missing at random (MAR) [23], but also provides reasonably stable results even when the assumption of MAR is violated [24,25]. Subjects who did not provide any diary data (leading to zero evaluable days) were excluded from the MAR-based primary efficacy analysis, according to an “all observed data approach” as proposed by previous authors [26]. This approach is statistically efficient without using multiple imputation techniques. Data retrieved after withdrawal of randomised study treatment were also included in the analysis.

Unless otherwise stated, all of the experimental results were expressed as mean ± SD of at least five replications. Statistical analysis of data was performed by the Student’s *t*-test or two-way ANOVA followed by the Tukey–Kramer multiple comparison test to evaluate significant differences between a pair of means. The statistic heterogeneity was assessed by using Cochran’s test (*p* < 0.1). The I2 statistic was also calculated, and I2 > 50% was considered as significant heterogeneity across studies. A random-effects model was used if significant heterogeneity was shown among the trials. Otherwise, results were obtained from a fixed-effects model. Percent change in mean and SD values were excluded when extracting SD values for an outcome. Standard deviation values were calculated from standard errors, 95% CIs, *p*-values, or *t* if they were not available directly. Previously defined subgroup analyses were performed to examine the possible sources of heterogeneity within these studies and included health status, study design, type of intervention, duration, total polyphenols dose, and Jadad score. Treatment effects were analysed using PROC MIXED with treatment and period as fixed factors, subject as random factor, and baseline measurements as covariates, and defined as weighted mean difference and 95% CIs calculated for net changes in faecal and serum parameters, and blood pressure values. Data that could not meet the criteria of variance homogeneity (Levenes test) and normal distribution (determined by residual plot examination and Shapiro–Wilks test) even after log transformation were analysed by a nonparametric test (Friedman). The level of significance (α-value) was 95% in all cases (*p* < 0.05).

#### 2.10.2. Analysis Sets

The full analysis set population included all randomised subjects, and subjects who did not fail to satisfy a major entry criterion. We excluded subjects who provided neither primary nor secondary efficacy data from efficacy analyses. The per protocol set consisted of all subjects who did not substantially deviate from the protocol; they had two characteristics. Firstly, this group included subjects for whom no major protocol violations were detected (for example, poor compliance, errors in treatment assignment). Secondly, they had to have been on treatment for at least 50 days counting from day of first intake (completion of a certain pre-specified minimal exposure to the treatment regimen). Hence, subjects who prematurely discontinued the study or treatment were excluded from the per protocol sample.

#### 2.10.3. Determination of Sample Size

In order to determine whether this study provided sufficient power to detect statistically significant differences using this design, a post-hoc power analysis was performed on the primary endpoints measured for all treatments (one-way repeated measures ANOVA, observed *F* = 0.6596, *α* = 0.05, 1 group, *n* = 20, 5 measurements, observed correlation among measurements = 0.7977, Geisser–Greenhouse sphericity *ε* = 0.4991). Based on these values, the statistical power was 100%, indicating that the sample size was sufficient to detect statistically significant differences if they were indeed present. This resulted in *n* = 20 subjects, which was increased to *n* = 30.

### 2.11. Subject Involvement

No subjects were involved in setting the research question or the outcome measures, nor were they involved in developing plans for participant recruitment, or the design and implementation of the study. There are no plans to explicitly involve subjects in dissemination. Final results will be sent to all participating sites.

## 3. Results

### 3.1. Polyphenolic Composition of AAP and lfAAP

Quantitative results from LC-MS analyses of polyphenols in AAP and lfAAP samples are reported in Table 1. Data show that fermentation of AAP for 24 h by using *L. rhamnosus* increased the amount of individual free polyphenols from a minimum of 20% up to 45% (total polyphenols: +32.7%), while at 48 h results were significantly lower (from a minimum of 10% up to 20%; total polyphenols: +14.9%), than non-fermented AAP, respectively. Conversely, fermentation of AAP using *L. plantarum* provided lower, but still significant results both at 24 and 48 h of incubation, than what was observed for AAP fermented using *L. rhamnosus*, respectively. As control, AAP samples were handled in the same incubation conditions (times and temperature) of lfAAP, but without the inoculation of any *Lactobacillus* strain: no significant variation of AAP polyphenolic compositions were registered. In the light of these results, lfAAP obtained by 24 h incubation with *L. rhamnosus* was selected as the best candidate sample to be administered to subjects.

### 3.2. Enrolment and Subject Attrition

Subjects were enrolled in January 2017. A total of 122 subjects were screened for eligibility; 32 subjects (26.2%) did not pass the screening stage; 90 subjects were randomised. The most common reason was that subjects did not meet the inclusion criteria regarding values of serum parameters at baseline (*n* = 14), followed by general refusal to participate for no specific reasons (*n* = 4), and concerns about the protocol, especially fear of placebo (*n* = 2). Some fulfilled exclusion criteria (*n* = 12). 

Overall, 90 subjects were assigned to the study. Subjects underwent a run-in period during the first four weeks before the treatment period of eight weeks. The follow-up period lasted another four weeks. Figure 1 shows the flow of participants through the trials together with the completeness of diary information over the entire treatment period. 

No subject prematurely terminated study participation before allocation to treatment. Figure 1 follows the CONSORT PRO reporting guideline [27] and reveals that within the assessment period, the following percentage of subjects provided data for the primary endpoint: subgroup AAP, 77.8% (21 of 27 subjects); subgroup lfAAP, 76.9% (20 of 26 subjects); subgroup LAB, 77.8% (21 of 27 subjects). In each group, a few subjects did not submit any diaries, giving no specific reason for this. Completeness of the subject diaries did not differ between the treatment groups.

### 3.3. Participants’ Baseline Characteristics

Table 2 shows the demographic and clinical characteristics assessed at the baseline visits of all 90 subjects randomised. Overall, about half of the randomised subjects were female; the total age range was 18–70 years. The groups were well balanced for demographics and clinical factors. 

### 3.4. Primary Efficacy Outcome Measures

No significant variations of plasma TC, LDL-C, HDL-C, glucose, TG, and TMAO levels, with respect to the baseline values, were registered in subjects of all subgroups, at the end of the run-in period (Table 2). Analysing results regarding the variation of plasma lipid values at the end of the intervention period (Table 3), we can assert that the most significant achievements were registered as regards HDL-C and TMAO levels in all subgroups. Specifically, the lfAAP sample exerted the highest influence on both parameters, increasing HDL-C levels by 61.8% (*p* = 0.0095) and lowering TMAO levels by 63.1% (*p* = 0.0042), at the end of the intervention period, followed by AAP sample (HDL-C: +48.4%, *p* = 0.0042; TMAO: −42.3%, *p* = 0.0032) and LAB sample (HDL-C: +17.7%, *p* = 0.0036; TMAO: −25.8%, *p* = 0.0029). It is noteworthy that all of the experimental results were achieved already after one month of intervention study and were confirmed at the end of the second month. Comparing the LC-MS results, regarding the effects of lactofermentation on the bioaccessibility of polyphenols in AAP samples (Table 1), with the clinical values, concerning the influence of lfAAP sample on plasma HDL-C and TMAO levels (Table 3), a significant correlation was found. Specifically, for the lfAAP sample exhibiting the highest bioaccessibility of polyphenols (Table 1), it was the most effective in increasing HDL-C levels and lowering TMAO levels in human plasma (Table 3). As regards the results at the end of the follow-up period, HDL-C and TMAO levels registered slightly significant variations in respect to the end of the treatment period (Table 3). As a general trend, a loss of efficacy was observed, as regards both parameters, at the end of the follow-up period, in respect to the end of the treatment period (t 60). The LAB treatment demonstrated the highest stability of efficacy (HDL-C, −6.6% (*p* = 0.0076); TMAO, −8% (*p* = 0.0058)), followed by the other two treatments ((lfAAP: HDL-C, −13.3% (*p* = 0.0061); TMAO, −12.9% (*p* = 0.0042); AAP: HDL-C, −10.4% (*p* = 0.0049); TMAO, −7.2% (*p* = 0.0037)).

### 3.5. Secondary Efficacy Outcome Measures

Results of the faecal swab microbial composition are reported in Table 4. On average, all of the samples (AAP, lfAAP, and LAB) determined a strong increase in *Bifidobacterium* and *Lactobacillus* population, and a slightly significant reduction of *Bacteroides* and *Enterococcus* genera, at the intestinal level. Very interestingly, AAP revealed the leading effects of increasing *Bifidobacterium* and *Lactobacillus* by about 74 and 300 times, respectively, followed by LAB (26 and 40 times, respectively), while lfAAP exerted a minor influence (3.5 and 2 times, respectively). Concerning all of these results, slightly significant variations, registered as a general decrease of efficacy, were revealed at the end of the follow-up period, respect to the end of the treatment period (Table 4). The lfAAP treatment demonstrated the highest stability of efficacy ((*Bifidobacterium*, −61.0% (*p* = 0.0055); *Lactobacillus*, −21.5% (*p* = 0.0079); *Bacteroides*, −9.0% (*p* = 0.0085); *Enterococcus*, −4.1% (*p* = 0.0034)), followed by the other two treatments (LAB: *Bifidobacterium*, −800% (*p* = 0.0028); *Lactobacillus*, −1050% (*p* = 0.0066); *Bacteroides*, −15.8% (*p* = 0.0043); *Enterococcus*, −14.8% (*p* = 0.0042); AAP: *Bifidobacterium*, −1460% (*p* = 0.0038); *Lactobacillus*, −12110% (*p* = 0.0053); *Bacteroides*, −15.0% (*p* = 0.0077); *Enterococcus*, −2.4% (*p* = 0.09)). 

### 3.6. Study Strength and Limitations

The major strengths of the clinical trial herein presented reside in the originality of the study and in the evaluation of the treatment effects in real-world settings. The positive results, herein reported, can inform physicians about a novel treatment/intervention, which can represent a valuable alternative in the clinical practice. Conversely, the main limitations of our study include the short-term assessment for the treatment of a chronic condition, the choice of exclusively white race, and the wide age range due to the availability of such individuals at the stage of the recruitment. Moreover, the lack of a placebo group, and the intention to make a treatment comparison, are clear indicators of the explorative nature of the present work, confirmed by the absence of a power analysis, which intends to define the bases of a more structured and statistically robust clinical study.

## 4. Discussion 

Our experimental results regarding the lactic acid fermentation of Annurca apple puree indicated a general increase of the polyphenolic compound levels in the final product, although this effect may strongly vary according to the type of *Lactobacillus* strain used and the incubation conditions (time and temperature) (Table 1). Polyphenols would be released from soluble and insoluble fibre during the fermentation process thanks to the action of bacterial hydrolases, thus, potentially enhancing their intestinal bioaccessibility [14,15,16]. Previous authors have demonstrated that ingestion of dietary polyphenols is followed by their absorption into the circulatory system through the small intestine [2]. However, it is well known that the bioavailability of polyphenols in the small intestine is quite low, mainly due to their general high polarity and molecular weight [2]. Nevertheless, a wide range of biochemical reactions occur along the entire intestinal tract which can significantly increase the possibility of polyphenols being absorbed: i) hydrolysis of glycosylated polyphenols by lactase and cytosolic β-glucosidases, releasing more lipophilic aglycones which may then enter the epithelial cells by passive diffusion [28]; ii) metabolism (sulfation, glucuronidation and/or methylation) of the same aglycones prior to passage into the blood stream [29]; and iii) further fermentation by colonic microflora, which can cleave conjugating moieties and subject the resultant aglycones to ring fission, leading to the production of phenolic acids and hydroxycinnamates, which can be absorbed in the large intestine [29]. All of these metabolites can be absorbed and ultimately excreted in urine in substantial quantities that, in most instances, are well in excess of the flavonoid metabolites that entered the circulatory system via the small intestine [2]. Therefore, it is evident that the increase in intestinal bioaccessibility of food polyphenols by lactic acid fermentation could lead to the formation of bioavailable antioxidant compounds with potential beneficial effects at systemic level.

Fermentation of AAP for 24 h using *L. rhamnosus* represented the best incubation conditions in terms of increase in polyphenol bioaccessibility (Table 1). Thus, this product (lfAAP) was chosen to be administered to subjects in order to evaluate its effects on plasma primary outcomes. Specifically, lfAAP exerted a significant influence on HDL-C levels, allowing a higher increase of its values than what was assessed in subjects assuming AAP and LAB (Table 3). Our research group previously evaluated the influence of daily consumption of Annurca apples on the cholesterol levels of mildly hypercholesterolemic healthy subjects [18].

Interestingly, a significant effect was also obtained by lfAAP on plasma TMAO values, decreasing its levels by 63.1% (*p* = 0.0042) in respect to what was revealed in subjects belonging to AAP (−42.3%; *p* = 0.0032) and LAB groups (−25.8%; *p* = 0.0029) (Table 3). Trimethylamine-N-oxide is currently recognized as a prognostic marker of oxidative stress for incident cardiovascular events beyond traditional risk factors [30,31]. Normal TMAO blood levels range from 0.5 to 5 µM [32,33], and increased TMAO serum levels are associated with a greater CVD risk [31]. Primarily, TMAO derives from the metabolism of choline and L-carnitine by the gut microbiota (mainly *Clostridia*, *Shigella*, *Proteus*, and *Aerobacter*), producing trimethylamine (TMA) which is oxidised by flavin-containing monooxygenase-3 (FMO3) in the liver [30,33,34,35,36]. In this context, the microbiota remodelling has been recognised as a novel and useful approach for the management of high TMAO levels-related diseases [37]. Particularly, evidence has shown that polyphenols are able to act as bactericide and bacteriostatic agents against some specific bacterial strains, mainly *Clostridia* and *Bacteroides* [38,39], and to favour the growth of non-TMA producing species, including *Prevotella* and *Firmicutes* [40], suggesting that a diet rich in polyphenols, or food supplements, may represent a strategy for microbiota remodelling. Actually, a further interpretation regarding plasma TMAO-reducing effect observed in this study may be formulated. Polyphenols are well-known as powerful antioxidant molecules, and it has been established that TMAO can act as an electron acceptor metabolite [41]. At the serum level, thus, TMAO and polyphenols may be involved in redox reactions, resulting in the generation of TMA. Very few recent studies have evaluated the effects of specific supplementations on plasma TMAO levels in humans. In the trial of Angiletta et al. [42], obese subjects at risk for insulin resistance, having elevated TMAO levels (≥5 μM) compared to levels previously measured in healthy subjects (∼1 μM), consumed tea or cocoa flavanols in a randomized crossover design while consuming a controlled diet [43]. Another study assessed the influence of inulin supplementation on plasma TMAO concentrations in individuals at risk for type 2 diabetes (TMAO levels ∼3 μM) [42]. Both studies concluded that the respective interventions did not significantly affect plasma TMAO levels in the enrolled subjects. Thus, our study is the first evidence of an effective dietary intervention on plasma TMAO levels in healthy subjects.

According to our secondary efficacy outcome measures, AAP was supposed to be the most effective in positively remodelling intestinal microbiota (Table 4). Although faecal swabs should be regarded as mere indicators of intestinal microbial population, and the four microbial genera (*Bifidobacterium*, *Lactobacillus*, *Bacteroides*, and *Enterococcus*), objects of our analyses are not fully representative of human gut microbiota, subjects assuming AAP revealed, in particular, a higher increase of fermentative bacteria than those administered with lfAAP. It could be hypothesised that AAP, richer in intact pectic compounds than lfAAP, may have provided subjects with substances of prebiotic interest which are well-known to strongly address the gut microflora balance in favour of the fermentative bacteria [44,45]. Furthermore, faecal swabs from subjects assuming AAP revealed a higher increase of fermentative bacteria even than those from LAB subgroup. Such results are of major interest since it would corroborate that prebiotics may play a major role in manipulating gut microbial composition with an efficacy even higher than what is possible through an attempt of direct gut colonisation with probiotics [46]. Undoubtedly, our results need to be deepened and supported by further experimental intervention.

## 5. Conclusions

The present study indicates lfAAP as an effective functional food for beneficial control of plasma HDL-C and TMAO levels, with clinical relevance in primary cardiovascular disease prevention. Obviously, further in vitro, in vivo, and human trials are needed to corroborate our results. Particularly, specific investigations will be necessary to identify the main constituents responsible for the clinical results observed in this study, and to evaluate their contribution to the described effects.

## Figures and Tables

**Figure 1 nutrients-11-00122-f001:**
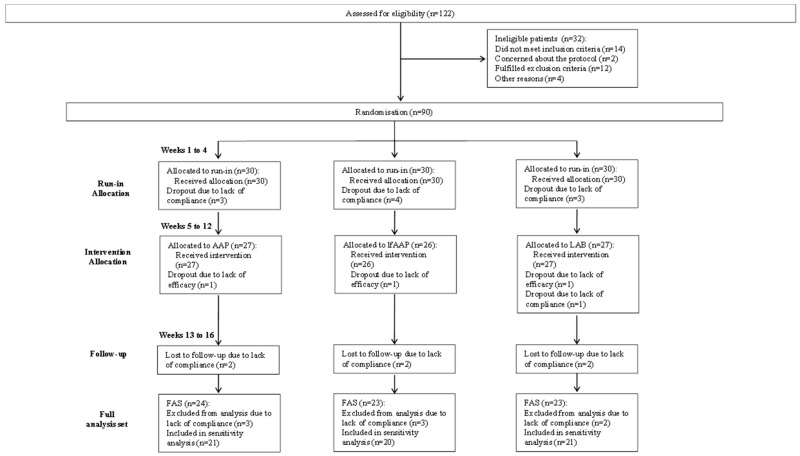
Study flowchart according to the consolidated standards of reporting trials (CONSORT). The diagram shows enrolment and primary efficacy endpoints based on subject diaries, from pre-screening to data collection; and the extent of exclusions, loss to follow-up, and completeness of diary documentation available across the entire trial period. AAP = subgroup administered with Annurca apple puree (125 g/day); lfAAP = subgroup administered with lactofermented Annurca apple puree (125 g/day); LAB = subgroup administered with 1 capsule/day, containing *Lactobacillus rhamnosus* (3.0 × 10^8^ CFU); FAS = full analysis set.

**Table 1 nutrients-11-00122-t001:** Polyphenolic composition of Annurca apple puree (AAP) and its lactofermented versions (lfAAP) obtained by incubation (at room temperature) with different *Lactobacillus* strains at different times.

	AAP	lfAAP (by *L. rhamnosus* at 24 h)	lfAAP (by *L. rhamnosus* at 48 h)	lfAAP (by *L. plantarum* at 24 h)	lfAAP (by *L. plantarum* at 48 h)
Chlorogenic acid	8.98 ± 0.02 ^a^	10.78 ± 0.03 ^b^	9.88 ± 0.02 ^c^	9.16 ± 0.04 ^a^	9.07 ± 0.05 ^a^
[+]-Catechin	1.20 ± 0.04 ^a^	1.52 ± 0.03 ^b^	1.34 ± 0.05 ^c^	1.25 ± 0.04 ^a^	1.22 ± 0.06 ^a^
[−]-Epicatechin	2.80 ± 0.06 ^a^	3.48 ± 0.05 ^b^	3.14 ± 0.04 ^c^	2.91 ± 0.03 ^a^	2.86 ± 0.07 ^a^
Procyanidin B1	0.70 ± 0.02 ^a^	1.01 ± 0.03 ^b^	0.84 ± 0.05 ^c^	0.78 ± 0.03 ^a^	0.75 ± 0.02 ^a^
Procianidin B2	1.78 ± 0.02 ^a^	2.58 ± 0.04 ^b^	2.14 ± 0.02 ^c^	1.99 ± 0.03 ^d^	1.88 ± 0.05 ^a,d^
Procyanidin trimer	1.28 ± 0.02 ^a^	1.87 ± 0.04 ^b^	1.54 ± 0.02 ^c^	1.43 ± 0.03 ^d^	1.36 ± 0.01 ^e^
Cyanidin-3-O-galactoside	0.04 ± 0.02 ^a^	0.05 ± 0.02 ^a^	0.05 ± 0.02 ^a^	0.04 ± 0.03 ^a^	0.04 ± 0.02 ^a^
Rutin (Quercetin-3-O-rutinoside)	0.80 ± 0.02 ^a^	1.08 ± 0.03 ^b^	0.93 ± 0.04 ^c^	0.86 ± 0.02 ^d^	0.83 ± 0.05 ^a^
Hyperin (Quercetin-3-O-galactoside)	8.90 ± 0.04 ^a^	12.1 ± 0.06 ^b^	10.3 ± 0.04 ^c^	9.61 ± 0.05 ^d^	9.26 ± 0.03 ^e^
Isoquercitrin (Quercetin-3-O-glucoside)	3.52 ± 0.02 ^a^	4.75 ± 0.03 ^b^	4.08 ± 0.06 ^c^	3.80 ± 0.05 ^d^	3.66 ± 0.03 ^e^
Reynoutrin (Quercetin-3-O-xyloside)	2.04 ± 0.08 ^a^	2.75 ± 0.05 ^b^	2.37 ± 0.07 ^c^	2.20 ± 0.06 ^d^	2.12 ± 0.08 ^a^
Guajaverin (Quercetin 3-O-arabinopyranoside)	1.74 ± 0.02 ^a^	2.35 ± 0.04 ^b^	2.02 ± 0.03 ^c^	1.88 ± 0.06 ^d^	1.81 ± 0.05 ^a^
Avicularin (Quercetin 3-O-arabinofuranoside)	3.96 ± 0.02 ^a^	5.35 ± 0.07 ^b^	4.59 ± 0.02 ^c^	4.28 ± 0.09 ^d^	4.12 ± 0.08 ^e^
Quercetin-O-pentoside	1.22 ± 0.02 ^a^	1.65 ± 0.04 ^b^	1.39 ± 0.03 ^c^	1.32 ± 0.02 ^d^	1.27 ± 0.05 ^a^
Quercitrin (Quercetin-3-O-rhamnoside)	2.34 ± 0.02 ^a^	3.16 ± 0.02 ^b^	2.71 ± 0.04 ^c^	2.53 ± 0.06 ^d^	2.43 ± 0.07 ^a^
Phloretin-2-O-xyloglucoside	2.70 ± 0.02 ^a^	3.78 ± 0.05 ^b^	3.19 ± 0.08 ^c^	2.97 ± 0.04 ^d^	2.83 ± 0.02 ^e^
Phloridzin (phloretin-2-O-glucoside)	3.02 ± 0.02 ^a^	4.23 ± 0.07 ^b^	3.56 ± 0.06 ^c^	3.32 ± 0.04 ^d^	3.17 ± 0.02 ^e^
Total polyphenols	47.02 ± 0.09 ^a^	62.40 ± 0.12 ^b^	54.07 ± 0.14 ^c^	50.33 ± 0.11 ^d^	46.25 ± 0.10 ^a^

Values are expressed as mg/125 g puree and are the means ± SD (*n* = 5; *p* < 0.01). ^abcde^ Mean values in rows with different superscript letters are significantly different by the Tukey–Kramer multiple comparison test. AAP = Annurca apple puree; lfAAP = lactofermented Annurca apple puree.

**Table 2 nutrients-11-00122-t002:** Baseline characteristics of intention to treat sample according to study treatment.

Run-in			
Characteristics	AAP (*n* = 30)	lfAAP (*n* = 30)	LAB (*n* = 30)
Demographics			
Age (years)	46.9 ± 10.6	45.8 ± 11.1	47.6 ± 10.4
Male sex (No (%))	17 (56.7%)	18 (60.0%)	16 (53.3%)
White ethnicity (No (%))	30 (100%)	30 (100%)	30 (100%)
Clinical parameters			
TC (mg/dL)	234.1 ± 13.2	238.5 ± 12.1	236.2 ± 11.8
LDL-C (mg/dL)	154.0 ± 11.1	155.7 ± 12.4	166.9 ± 12.0
HDL-C (mg/dL)	37.8 ± 6.3	38.3 ± 7.4	40.5 ± 6.6
Glucose (mg/dL)	100.5 ± 8.2	102.2 ± 9.3	110.2 ± 9.1
Triglycerides (mg/dL)	178.1 ± 9.6	188.1 ± 11.7	197.6 ± 11.6
TMAO (μM)	2.01 ± 0.05	2.68 ± 0.06	1.72 ± 0.06
**Treatment**			
**Characteristics**	**AAP (*n* = 27)**	**lfAAP (*n* = 26)**	**LAB (*n* = 27)**
Demographics			
Age (years)	45.1 ± 10.3	46.2 ± 10.7	48.2 ± 10.2
Male sex (No (%))	15 (55.5%)	16 (61.5%)	15 (55.5%)
White ethnicity (No (%))	27 (100%)	26 (100%)	27 (100%)
Clinical parameters			
TC (mg/dL)	235.5 ± 13.3	237.6 ± 14.3	239.1 ± 11.9
LDL-C (mg/dL)	152.1 ± 11.1	156.4 ± 11.6	165.8 ± 11.7
HDL-C (mg/dL)	36.7 ± 7.5	37.4 ± 6.7	38.5 ± 7.2
Glucose (mg/dL)	99.1 ± 2.7	100.0 ± 8.9	107.2 ± 8.5
Triglycerides (mg/dL)	180.4 ± 16.8	185.2 ± 18.3	200.1 ± 19.0
TMAO (μM)	2.37 ± 0.04	3.01 ± 0.05	2.02 ± 0.06

Values are means ± SD (*n* = 5). Results were significantly different at a level of *p* = 0.001. AAP = subgroup to be administered with Annurca apple puree; lfAAP = subgroup to be administered with lactofermented Annurca apple puree; LAB = subgroup to be administered with *Lactobacillus rhamnosus*. TC: total cholesterol.

**Table 3 nutrients-11-00122-t003:** Effects of Annurca apple puree (AAP) and its lactofermented version (lfAAP) on plasma cholesterol, glucose, triglyceride, and TMAO levels.

		AAP	Δ (%)	lfAAP	Δ (%)	LAB	Δ (%)
TC (mg/dL)	t 0	235.5 ± 13.3		237.6 ± 14.3		239.1 ± 11.9	
	t 30	246.1 ± 14.2	+4.5	257.1 ± 12.8	+8.2	242.2 ± 13.5	+1.3
	t 60	246.8 ± 13.6	+4.8	259.0 ± 12.1	+9.0	243.4 ± 13.8	+1.8
	t 90	248.2 ± 14.0	+5.4	259.5 ± 16.7	+9.2	253.1 ± 14.2	+5.8
LDL-C (mg/dL)	t 0	152.1 ± 11.1		156.4 ± 11.6		165.8 ± 11.7	
	t 30	157.3 ± 11.3	+2.7	163.5 ± 10.9	+3.6	168.7 ± 10.8	+1.5
	t 60	158.6 ± 13.6	+3.4	164.8 ± 11.2	+4.3	169.7 ± 11.2	+2.0
	t 90	160.0 ± 12.5	+5.2	166.2 ± 11.1	+6.3	172.6 ± 10.7	+4.1
HDL-C (mg/dL)	t 0	36.7 ± 7.5		35.4 ± 6.7		38.5 ± 7.2	
	t 30	53.6 ± 7.5 *	+46.1 ^#^	56.2 ± 7.9 *	+58.9 ^#^	44.3 ± 8.0 *	+15.2 ^#^
	t 60	54.5 ± 7.0 *	+48.4 ^#^	57.3 ± 8.1 *	+61.8 ^#^	45.3 ± 8.3 *	+17.7 ^#^
	t 90	49.6 ± 6.8 *	+35.1 ^#^	53.6 ± 7.5 *	+51.4 ^#^	42.8 ± 7.6 *	+11.1 ^#^
Glucose (mg/dL)	t 0	99.1 ± 8.7		100.0 ± 8.9		107.2 ± 8.5	
	t 30	102.0 ± 10.8	+2.9	103.8 ± 12.0	+3.8	109.4 ± 12.4	+2.1
	t 60	102.5 ± 11.6	+3.4	104.5 ± 13.1	+4.5	109.9 ± 10.3	+2.5
	t 90	103.0 ± 12.1	+3.9	105.1 ± 12.4	+5.1	110.2 ± 11.9	+2.8
Triglycerides (mg/dL)	t 0	180.4 ± 16.8		185.2 ± 18.3		200.1 ± 19.0	
	t 30	185.1 ± 19.3	+2.6	192.2 ± 16.4	+3.8	203.3 ± 21.1	+1.6
	t 60	186.2 ± 14.7	+3.2	193.5 ± 17.3	+4.5	204.5 ± 15.2	+2.2
	t 90	189.0 ± 16.2	+4.8	194.6 ± 17.0	+5.1	206.9 ± 18.7	+3.4
TMAO (μM)	t 0	2.37 ± 0.04		3.01 ± 0.05		2.02 ± 0.06	
	t 30	1.45 ± 0.02 *	−38.9 ^#^	1.24 ± 0.06 *	−58.7 ^#^	1.56 ± 0.07 *	−22.9 ^#^
	t 60	1.36 ± 0.05 *	−42.3 ^#^	1.11 ± 0.04 *	−63.1 ^#^	1.49 ± 0.04 *	−25.8 ^#^
	t 90	1.54 ± 0.04 *	−35.1 ^#^	1.50 ± 0.06 *	−50.2 ^#^	1.66 ± 0.04 *	−17.8 ^#^

AAP = subgroup administered with Annurca apple puree (125 g/day); lfAAP = subgroup administered with lactofermented Annurca apple puree (125 g/day); LAB = subgroup administered with 1 capsule/day, containing *Lactobacillus rhamnosus* (3.0 × 10^8^ CFU). *: Significantly different from baseline at *p* < 0.01 (PROC MIXED). #: Significantly different from the other treatments at *p* < 0.05 (PROC MIXED).

**Table 4 nutrients-11-00122-t004:** Effects of Annurca apple puree (AAP) and its lactofermented version (lfAAP) on intestinal microbial composition.

		AAP	Δ (%)	lfAAP	Δ (%)	LAB	Δ (%)
Bifidobacterium (CFU/mL)	t 0	10,499 ± 2458		20,341 ± 4316		15,239 ± 3104	
	t 30	637,289 ± 82,567 *	+6970 ^#^	62,040 ± 5018 *	+205 ^#^	335,258 ± 45,128 *	+2100 ^#^
	t 60	776,961 ± 74,329 *	+7300 ^#^	70,379 ± 6894 *	+246 ^#^	396,214 ± 45,219 *	+2500 ^#^
	t 90	623,640 ± 77,241 *	+5840 ^#^	57,971 ± 5746 *	+185 ^#^	274,302 ± 39,489 *	+1700 ^#^
Lactobacillus (CFU/mL)	t 0	3584 ± 532		2989 ± 387		3467 ± 478	
	t 30	748,339 ± 64,651 *	+20,780 ^#^	6127 ± 597 *	+105 ^#^	124,812 ± 49,731 *	+3500 ^#^
	t 60	1,078,784 ± 109,657 *	+30,000 ^#^	6486 ± 529 *	+117 ^#^	140,413 ± 14,521 *	+3950 ^#^
	t 90	644,761 ± 58,452 *	+17,890 ^#^	5843 ± 509 *	+95.5 ^#^	104,010 ± 23,464 *	+2900 ^#^
Bacteroides (CFU/mL)	t 0	7843 ± 755		5671 ± 487		6891 ± 337	
	t 30	3529 ± 550 *	−55.1 ^#^	4139 ± 364 *	−27.5 ^#^	5168 ± 499 *	−25.8 ^#^
	t 60	2901 ± 351 *	−63.2 ^#^	3856 ± 405 *	−32.1 ^#^	4410 ± 461 *	−36.0 ^#^
	t 90	4062 ± 259 *	−48.2 ^#^	4360 ± 422 *	−23.1 ^#^	5499 ± 415 *	−20.2 ^#^
Enterococcus (CFU/mL)	t 0	6914 ± 419		8317 ± 575		7819 ± 561	
	t 30	6706 ± 731	−3.7 ^#^	7651 ± 645 *	−8.4 ^#^	5082 ± 484 *	−35.5 ^#^
	t 60	6568 ± 457	−5.3 ^#^	7402 ± 534 *	−11.6 ^#^	4456 ± 379 *	−43.0 ^#^
	t 90	6713 ± 610	−2.9 ^#^	7693 ± 625 *	−7.5 ^#^	5614 ± 404 *	−28.2 ^#^

Data are expressed as CFU/mL of faecal swab sample. Values are means ± SD (*n* = 5). * Significantly different from baseline at *p* < 0.01 (PROC MIXED). ^#^ Significantly different from the other treatments at *p* < 0.05 (PROC MIXED). t 0: 1st day of treatment; t 30: 30th day of treatment; t 60: 60th day of treatment; t 90: 30th day of follow-up. AAP = subgroup administered with Annurca apple puree (125 g/day); lfAAP = subgroup administered with lactofermented Annurca apple puree (125 g/day); LAB = subgroup administered with 1 capsule/day, containing *Lactobacillus rhamnosus* (3.0 × 10^8^ CFU).

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
