# Peer review of "Lactofermented Annurca Apple Puree as a Functional Food Indicated for the Control of Plasma Lipid and Oxidative Amine Levels: Results from a Randomised Clinical Trial"

_nutrients, 2019, doi:10.3390/nu11010122_

Reviewer 1 Report

Annurca apples are highly enriched in polyphenols. Supplementation of 2 annurca apples was shown to increase HDL-C by around 15%. Fermentation of Annurca apples could potentially increase the bioavailability of the polyphenols so the authors set out to see if the supplementation of fermented Annurca apple puree (125g) for 8-weeks could improve blood lipids (HDL-C)

1) The authors noted a significant increase in HDL-C in all treatment groups. The most dramatic increase was observed in the lfAAP group with a slightly lower increase in the AAP group. The authors did not note whether the HDL-C increase in the lfAAP group was statistically different than the AAP group This is important in the context of framing the rest of their story that suggests that the increased bioavailability of the polyphenols by approximately 33% based on table 1 can improve HDL-C and TMAO. This reviewer would like to know if they are indeed significant. The authors seem to infer that there is a significant correlation (lines 359-364), but the data is not shown (If they aren't, the authors conclusions are not valid. Why was the increase in HDL-C so much larger than what was observed in their previous study using 2 apples (~ 250g/day)?

2) The methods section is scant on the description of how the AAP, lfAAP, and LAP are given. Are the study subjects provided all 8-weeks of the treatment doses at randomization? Are they instructed to eat it at a certain time of the day? Do they eat it with food? All at one time? Is there any preparation required (i.e. is it lyophilized)? Is it given as pills? Is LAP given as a capsule? Does LAP look exactly the same as AAP and lfAAP? If not, this is not a double blind study.

3) How did the authors choose the inclusion criteria for the study? The abstract refers to the subjects as "healthy subjects". The inclusion criteria for the blood lipids are not in the range that would be classified as "normal".

4) Is there a typographical error in the inclusion criteria for LDL-C? The methods section notes 189-206mg/dl, but all of the subjects seems to fail that inclusion criteria based on Table 3.

5) The authors refer to the study volunteers as patients throughout the text of this manuscript. Please note that study subjects are not patients. The researchers are not managing the medical care of these individuals. Please refer to the study volunteers as study subjects.

6) Do the authors have a proposed mechanism for the HDL-C increase? Two large randomized control trials of niacin supplementation, AIM-HIGH and HPS2-TRIVE, have shown that despite significant increases in HDL-C, niacin does not decrease the incidence of cardiovascular events. Do the authors suggest that the HDL-C increase from AAP will decrease CVD risk?

7) The authors use the term "oxidative stress" in the title. The authors did not evaluate oxidative stress with traditional markers such as TBARS, malondialdehyde, nitrotyrosine, or isoprostinoids. The decrease in TMAO does not necessarily mean that there is decreased oxidative stress. If the authors would like to use that term in the manuscript title, please include data that supports the reduction.

8) Line 125: Lyophilised should be lyophilized

9) It is unclear what analysis was done by PDA and which were done by LC-MS. Can you please clarify? It appears that you verified the standard purity by LC-MS. Why did you not quantify by LC-MS?

10) Line 196; Centrifugal force should be reported as XG or RCF, not by RPM.

11) Line 199: "For the identification (ESI) source operated in positive ion mode." Please clarify this statement

12) Lines 241-242: The decision process was performed according to a consensus document (unpublished 241 standard operating procedure) . Please describe this procedure in more detail.

13) Line 305: Which results are quantitative and which are quantitative? 

14) Lines 189-193: Which concentrations were used to define the assay sensitivity?

15) Are any of the changes in the microbial compositions significantly different? If not, the comments in the results section regarding the size of the increase are not valid and the statement in line 449 needs to be removed. Why are the authors presenting the data in Table 4 in log CFU? The legend below reads "CFU/mL" which is discordant from the data reported above. The log transformed numbers can be confusing to readers which may misinterprete the true change in colony forming units if the changes are indeed statistically significant.

19) Please remove the word "proved" in line 389. Under the experimental conditions tested, the authors observed an increase, which supports their hypothesis. In fact, longer periods of fermentation was not supportive of elevated polyphenol levels. The authors do not describe why there is a decrease in polyphenol levels with prolonged fermentation. Are the bacteria metabolizing the polyphenols into other metabolites that are not being analyzed by their methods? Are these molecules bioactive?

19) Prodromal is not the correct word choice.

Author Response

Authors are very grateful for the Reviewer’s comments. In accordance to the Reviewer’s indications, corrections/integrations have been made to the manuscript and are reported in the attached file.

Reviewer 2 Report

This is an interesting study testing the effects of Apple puree, fermented apple puree, and probiotic on lipids and TMAO.

The abstract is concise, but almost to concise. The reviewer did not understand that the study had 3 treatments and randomized so many people. It was also unclear that there was no true negative control group. All treatments were active, and based on review of the data this indeed was true.

This is a very elegant design (4 weeks run-in, 8 weeks treatment, 4-week off/follow-up). There was nearly a ½ page plus about the statistics which seemed to be quite complicated, when in the end it appeared no treatment comparisons were made. The sample size estimates were not performed a priori, but instead a post-hoc analysis of power was conducted. The power as based on within treatment change (I think) and not between treatment differences. In general, a statistical expert should review what the authors did. It was quite confusing, given what appeared to be a straight forward analysis (using Intent to treat and or completer dataset).

The authors state “the aims of the present work were to: i) evaluate the effects of lactic acid fermentation on the polyphenolic bioaccessibility of Annurca apple puree; ii) test a food product based on lactofermented Annurca apple puree on humans in order to evaluate its effects on plasma lipid profile and oxidative stress.”

The hypothesis was not stated for which statistics would be based.

Primary endpoints – long list. Primary infers “1” endpoint as primary. Why not one primary for hypothesis testing and then rest secondary or tertiary depending on study objectives?

Apart from not having a true negative control, all treatments active to some degree, and not statistically analyzing for differences between treatments, the major missing part of this study was the lack of analysis of polyphenols and their metabolites in blood. This would give a much better indication of the importance of “bioaccessibility” and what was more bioavailable?

A paragraph in discussion on study limitations not include. Please include.

I/E – This reviewer was surprised how many people met all the criteria lipid criteria, especially since in some countries the people would be eligible for medication. Subjects were also not obese, which was surprising given the lipoid and glucose parameters.

This is the first time I have seen HDL increase this much with food-based or dietary-based intervention. Might be worth having a discussion about these results compared to other dietary regimens.  

Why did the authors choose to only report days 0, 30, 60? Why not all these days plus the day 90, which was 30 day after intervention completed? Please add these data.

Author Response

Authors are very grateful for the Reviewer’s comments. The points raised by the Reviewer have been addressed as reported in the attached file.

Round  2

Reviewer 1 Report

This reviewer thanks the authors for their revisions and their comments addressing my critiques. The manuscript is significantly improved in its current form. I only have 5 additional comments that should be addressed prior to publication.

 1)   Lines 447-452 state “Two Annurca apples a day for two months allowed an increase of HDL-C levels by 15.2% (P < 0.001) at the end of the intervention period. Comparing the weight of two Annurca apples (about 2 x 100g) with that of the daily portion of lfAAP (125g) assumed by subjects, it is hypothesisable that the more important effect on HDL-C levels revealed in the lfAAP group may be ascribed to the higher aliquot of bioacessible polyphenols assumed by subjects.”

a.    Please remove this statement. This reviewer is now in agreement based on the data presented in this manuscript that fermentation of apple puree for 24 hours is associated with increased polyphenol levels in the apple products and that treatment of subjects with this fermented product is associated with a significant increase in HDL-C levels relative to apple puree (non-fermented) or the bacteria used to ferment the apple. However, the comparison of the lfAAP group in this study with the apple group in the previous study is not valid because the apple puree (AAP) group, which is the most direct comparison to the previous study, received only 125g of apple products compared to 200g in the previous study, and was associated with a 48.4% increase in HDL-C over the two-month study period. This is much higher than the aforementioned 15.2% in the previous study.

b.    “Assumed” should be “consumed”

2)   “As stated at lines 188-193, all of the subjects were provided all 8-weeks of the treatment doses at randomization.”

a.    This is not explicitly stated in lines 188-193. Please add this verbiage.

3)   However, authors are ready to correct the definition of the clinical trial design by deleting the words “double-blind”, in case the Reviewer would disagree with the authors’ reply.

a.    Please remove the words “double-blind”

4)   Qualitative results refer to the characterization of individual polyphenols, while quantitative results refer to their concentration in the puree samples (expressed as mg/125 g puree).

a.    The analysis presented in table 1 is quantitative. Please change line 335 to read “Quantitative results from LC-MS”…

5)   There appears to be a discrepancy in what is written in lines 338-340 and table 1.

a.    48 hours of incubation with L. rhamnosus are significantly lower than 24 hours, but are still signifincatly higher than AAP (no fermentation)

b.    Please remove “slightly significant”. Results are either statistically significant or not statistically significant. (line 339)

c.    24 hours of incubation with L.  plantarum is associated with a significant increase in polyphenols, however 48 hours is not relative to no fermentation. The text reads “fermentation of AAP by using L. plantarum provided slightly significant results both at 24 and 48 hours. Please remove “slightly” and “48”.

6)   Previous literature suggests that increased polyphenol bioavailability is associated with increased systemic (plasma) levels.  While this is most likely true in this study, that data is not presented in this manuscript. That assumption should be explicitly described in the discussion.

7)   Typos

a.    Line 56 not bioaccesible should be non-bioaccessible.

b.    Line 194: change one of the daily main meal to one of the meals

Author Response

The authors are very grateful for the Reviewer's help. A detailed response list is in attachment.

Reviewer 2 Report

Authors did not address reviewer comments about the abstract.

The abstract does not adequately describe the study. The design information suggests it was double-blind and placebo controlled. There was no placebo. How was the double blind implemented? Abstract only talks about 26 subjects. 

Consider re-writing to be clear, ie., Ninety individuals with CVD risk factors were randomized (1:1:1 ratio) to a 3 arm, 8 week intervention and 4 week follow up testing the effect of lactofermtented Apple puree compared to unfermented Apple puree or Probiotic alone on plasma lipids, XX, XX. Results written as if within treatment comparison not between treatment comparison. This needs to be clear. Abstract conclusions are over-stating. 

What is meant by "preliminary results" in title? 

The lack of a priori power analysis and long list of "primary" endpoints suggests this is an exploratory study. If so, this should be stated and for the purposes of ......

The authors overstate how these data can be used in clinical practice the strengths and limitations section. This section needs to expand to include the lack of placebo in the study. All arms were active.

In the introduction... seems that the study objectives may need to be re-written if the aim (as it seems) is to determine if fermenting apple puree will improve the efficacy profile of the apple puree, and in doing so, control for the probiotic component in a parallel arm.

Results section... vague. After 30 day follow up there was slight variation in results. This should be made clear. Was there a significant increase or decrease from t 60? Were the values back to baseline t90 vs t0. How did the values compare between treatments?

What is meant by "heathy control" of HDL or "healthy control" of TMAO? 

Author Response

(The authors gave the same response as above.)
